# The Potential for Sialic Acid and Sialylated Glycoconjugates as Feed Additives to Enhance Pig Health and Production

**DOI:** 10.3390/ani11082318

**Published:** 2021-08-05

**Authors:** Marefa Jahan, Nidhish Francis, Peter Wynn, Bing Wang

**Affiliations:** Graham Centre for Agricultural Innovation, School of Agricultural, Environmental and Veterinary Sciences, Charles Sturt University, Locked Bag 588, Wagga Wagga, NSW 2678, Australia; mjahan@csu.edu.au (M.J.); nfrancis@csu.edu.au (N.F.); pwynn@csu.edu.au (P.W.)

**Keywords:** sialic acids, oligosaccharide, lactoferrin, ganglioside, pig, health, production

## Abstract

**Simple Summary:**

This review discusses the current challenges in the pig industry and the potential nutritional significance of sialic acid (Sia) and glycoconjugates (Sia-GC’s) for pig health and nutrition. Sia is a nine-carbon acidic sugar which is present in various organs and body fluids of humans and animals. Sias contribute to many beneficial biological functions including pathogen resistance, immunomodulation, gut microbiota development, gut maturation, anti-inflammation and neurodevelopment. The role of Sias in regulating the metabolism of pigs has seldom been reported. However, we have documented significant beneficial effects of specific Sia-GC’s on health and production performance of sows and piglets. These findings are reviewed in relation to other studies while noting the beneficial effects of the inclusion of Sia, Sia containing oligosaccharide or the sialo-protein lactoferrin in the diets of gilts and sows. The importance of the passive transfer of of Sia and Sia-GC’s through milk to the young and the implications for their growth and development is also reviewed. This information will assist in optimizing the composition of sow/gilt milk replacers designed to increases the survival of IUGR piglets or piglets with dams suffering from agalactia, a common problem in pig production systems worldwide.

**Abstract:**

Swine are one of the most important agricultural species for human food production. Given the significant disease challenges confronting commercial pig farming systems, introduction of a new feed additive that can enhance animal performance by improving growth and immune status represents a major opportunity. One such candidate is sialic acid (Sia), a diverse family of nine-carbon acidic sugar, present in various organs and body fluid, as well as an essential structural and functional constituent of brain ganglioside of humans and animals. Sias are key monosaccharide and biomarker of sialylated milk oligosaccharide (Sia-MOS’s), sialylated glycoproteins and glycolipids in milk and all vertebrate cells. Sias accomplish many critical endogenous functions by virtue of their physiochemical properties and via recognition by intrinsic receptors. Human milk sialylated glycoconjugates (Sia-GC’s) are bioactive compounds known to act as prebiotics that promote gut microbiota development, gut maturation, pathogen resistance, immunomodulation, anti-inflammation and neurodevelopment. However, the importance of Sia in pig health, especially in the growth, development, immunity of developing piglet and in pig production remains unknown. This review aims to critically discuss the current status of knowledge of the biology and nutritional role of Sia and Sia-GC’s on health of both female sow and newborn piglets.

## 1. Introduction

In the swine industry, several factors including neonatal mortality, weaner ill-thrift, infectious and non-infectious diseases can negatively affect pig populations worldwide. In addition, improvements in management practices, disease control and vaccination programs, nutritional supplementation is an effective strategy that can be implemented to improve pig production. Most nutritional strategies have focused on the requirements for specific nutrients in the diet to promote maximum growth. In particular, recommendations for vitamins and minerals have been at the forefront of porcine nutritional research. More recently, nutritional intervention has focused on several compounds that may act through multiple ways to boost the immune response thereby prevent or delay the occurrence of diseases in pigs.

One group of nutrients, which have attracted significant attention in recent years is the complex functional carbohydrates, more specifically sialic acid (Sia). Sia is a nine-carbon backbone acidic sugars decorating all cell surfaces and is one of the most secreted proteins of vertebrates. Sia has drawn the attention of many researchers due to its diverse beneficial biological functions through mediating or modulating a variety of normal and pathological processes both in humans and animals. Porcine milk is a rich source of Sia for the piglets. This review gives an overview of current challenges in the pig industry and the potential nutritional significance of Sia and sialylated compounds for pig health and pig industry.

## 2. Problems of the Pig Industry

One of the major issues faced by pig farmers is the frequent incidence of intrauterine growth restricted (IUGR) piglets. Notably, the pig as an animal with multifetal pregnancies, exhibits the most severe naturally occurring IUGR amongst domestic animals due to placental insufficiency [1]. Approximately 15–20% of piglets have a birth weight ≤1.1 kg compared with a normal birth weight of 1.4 kg [2]. A variety of physiological and production-imposed conditions are responsible for IUGR in pig industry [3]. IUGR increases piglet morbidity and mortality during the early postnatal life. IUGR also has a permanent stunting effect in postnatal growth and hinders the efficiency of nutrient utilization in offspring. Moreover, IUGR negatively affects whole-body composition and meat quality and impairs long term health [2]. In addition, IUGR has been found to be associated with immunodeficiency and increased vulnerability to infectious diseases in later life [4,5,6].

Pre-weaning mortality (PWM), a measure of the number of piglets that did not survive during the suckling period, remains a major welfare and economic problem in pig industries. The mean piglet PWM rate in commercial pig herds ranges between 10% and 20% in major pig-producing countries [7]. Specifically, the report showed a mean piglet PWM rate of 12.9% in the European Union (EU), 9.4% in the Philippines and 12.2% in Thailand [7].

Infectious diseases affecting pigs is a major economic loss to the industry with a substantial increase in the morbidity and mortality rate. The aetiology of these infectious diseases are primarily bacterial, viral or parasitic in origin and these pathogens vary worldwide leading to shift in the priorities of research on pig diseases in the last two decades. An extensive review outlines the global trends of infectious diseases in pigs where the authors reviewed >57,000 publications from 1996 till 2016 [8] and reported that the most common viral infections in pigs include influenza, foot and mouth disease, porcine reproductive and respiratory syndrome and pseudorabies (Aujeszky’s disease). Bacteria, most commonly reported to infect pigs included *Salmonella*, *Escherichia coli*, *Actinobacillus pleuropneumonia* and *Pasteurella multocida* and the most reported parasites were *Ascaris suum* and *Trichella* [8].

When delineating the potential strategies to fight against these problems of the pig industry, it is worth noting that primiparous sows (gilts) are very much more prone to many of these aforementioned issues than multiparous sows. For instance, it has been shown that gilts normally produce higher rate of IUGR piglets compared with mature sows [9]. The piglets born from gilts have reduced weight, slower growth rate and increased mortality rate [10]. Additionally, postparturient disorders including abnormal vaginal discharge, fever and reduced appetite and backfat loss were significantly greater in gilts than multiparous sows [11] implying the need of additional postpartum care and attention to gilts. One approach to combat these issues is through nutritional supplementation with bioactive compounds that will act at nutraceutical and therapeutical agent to improve the physiology, immunity and overall general health status of the pig.

## 3. Sialic Acid and Its Diversity in Nature

A crystalline reducing acid from bovine submandibular mucin was isolated by Gunnar Blix in 1936 [12]; thereafter it was named ‘sialic acid’ (sialos is the Greek word for Saliva) [13]. The term Sia is preferably used for nine-carbon acidic amino sugars, which are based on neuraminic acid (Neu). They are usually found as a α-glycoside commonly occupying the non-reducing end of hetero oligosaccharides in glycoconjugates, such as glycoproteins and glycolipids. There are more than 50 members of the Sia family including *N*- and O-substituted Sia [14]. The most common molecular species found in glycoproteins and glycolipids are *N* acetylneuraminic acid (Neu5Ac, NANA) and *N*-glycolylneuraminic acid (Neu5Gc), which are found on the cell surface and secreted glycoconjugates of vertebrates, so-called “higher” invertebrate animals [15,16,17,18]. 3-deoxy-d-glycero-d-galacto-nonulosonic acid (KDN) is a naturally occurring deaminated neuraminic acid and found more frequently at the nonreducing ends of oligosialyl chains in polysialoglycoprotein in lower vertebrates, including the sperm, the eggs of the rainbow trout and teleost fish [19,20,21,22]. Neu5Ac is believed to be the biosynthetic precursor for all other members of the family [20]. However, Neu5Gc is thought to be absent in healthy humans [23] and white meat, e.g., chickens [24], but is best investigated in porcine tissue [25]. Sias are generally not found in plants, prokaryotes or invertebrates [26], but some pathogenic bacteria contain Sias, which are thought to be “acquired” from hosts by gene transfer [25,27]. Later on, it was found that the relevant genes of bacterial pathogens were only distantly homologous to corresponding host genes [15]. Types of nonulosonic acids which are sometimes designated as “bacterial Sia” are legionaminic acid and pseudaminic acid [28].

Although there is a limited subset of nonulosonic acids in the deuterostome lineage of animals, there are considerable types of natural modification compared to any other common monosaccharides [15,25,29]. The reasons for this chemical diversity are not entirely clear, but it is hypothesized that they are the outcome of ongoing evolutionary selection by host–pathogen interactions to conserve critical endogenous functions [15,29]. Most of the diversity arises from modifications at C-5, e.g., the *N*-glycolyl group, or at the C-4, C-7, C-8 and C-9 hydroxyl groups, e.g., *O*-acetyl esters. Additional diversity in Sia molecule arises from a variety of glycosidic linkages (introduction of double bond) from C-2 to the underlying glycans, which gives rise to a wide variety of isomers [30]. Further complexity arises from the fact that these linkages can be presented on different underlying glycan chains. An unusual modification is an additional hydroxyl group at position 5 of the sugar, leading to a 2-keto-3-deoxy-d-glycero-d-galacto-nononic acid (ketodeoxynonulosonic acid, KDN) (Figure 1). All these modifications can further alter the biology of Sias. A further level of complexity arises from the fact that, sialoglycans can be organized into “clustered saccharide patches” on cell surfaces that involve interactions with other glycans, modulating recognition by different Sia-recognizing proteins [31].

## 4. Biological Functions of Sialic Acids

Sia are ubiquitous on all vertebrate cell surfaces (also in certain pathogenic/commensal bacteria) and are essential glycans for embryonic development [32]. They accomplish many critical endogenous functions by virtue of their physical properties and via recognition by intrinsic receptors [16,33] in modulating interactions with the environment. The negative charge and hydrophilicity of Sia helps in the biophysical features of many cells in the body [34]. One of the extensively studied biological roles of Sia is related to gut microbiota colonization, development, gut maturation, immunomodulation, pathogen resistance, anti-inflammation and neurodevelopment (Figure 2)

### 4.1. Development of Gut Microbiota

During the early life of a neonate, the gut microbiota is relatively simple in composition, but the abundances and diversity of gut microbiota have undergone dynamic changes during animal maturation. The colonization of the neonatal gut is influenced by many factors including maternal microbiota, breastfeeding versus formula feeding and environmental exposure [35]. Sia is the key molecular unit of sialylated milk oligosaccharides (Sia-MOS’s), which are also able to shape the gut microbial communities [36], as different milk oligosaccharides are fermented by different microbiota in the gut [37]. Eventually, the growth and metabolism of *Bifidobacterium* and *Bacteroides* strains are stimulated by the presence Sia-MOS’s [38], which may subsequently protect the newborn against infectious pathogens [39,40]. Sialyllactose (SL) has been revealed to support not only the growth of *B. longum* subsp. *infantis* but also its adhesion to intestinal epithelial cells, which may be associated with intestinal colonization [41]. Moreover, an infant specific *Bifidobacterium infantis* possesses fucosidase and sialidase activities, which are not present in several other *Bifidobacteria* strains [42,43]. All these *Bifidobacteria* and *Bacteroides* eventually take part in the fermentation of oligosaccharides resulting in the production of short-chain fatty acids (SCFAs) [44]. Gut microbiota and SCFAs play a very important role in the protection against pathogens and gut maturation. SCFAs influence nutrient absorption, host metabolism (amino acid, lipid, antioxidant and drug) and immune function, in addition, to their anti-diarrheal function by stimulating absorption of water and sodium [45]. Furthermore, SCFAs have been involved in reducing risk of diseases, such as cardiovascular disease, cancer and inflammatory bowel disease [46].

### 4.2. Gut Maturation and Gut Pathogen Resistance

Sia is believed to influence gut maturation in early life. In neonatal rat, Sia on the surface of the intestinal microvilli has been shown to boost the binding of IgG antibodies to the epithelium [47]. Towards the time of weaning, after which maternal IgG is no longer available, the expression of Sia decreased, which is explained by a decrease in α-2,6-sialyltransferase activity [48]. Sia-MOS’s may inhibit intestinal epithelial cell proliferation and promote cell differentiation [49,50]. These outcomes are mediated via interaction of the acidic oligosaccharides with carbohydrate moieties on the epidermal growth factor (EGF) receptor, which then regulates the activation of cell differentiation in the intestinal epithelial cell [50]. By modulation of EGF receptor signaling, EGF and Sia-MOS’s may promote intestinal maturation in early life [51].

Sia as glycoconjugates are present on bacterial surfaces as well as on host cell membranes. Sia has multiple substitutional binding sites and is capable of binding to adjacent molecules through distinct glycosidic linkages [52]. By competing for these binding sites, Sia components, can prevent or reduce adhesion of pathogens to host cell membranes. The ability of Sia-MOS’s to protect against infectious agents may result, in part, from their effects on the gut microbiota, but it is thought to be due primarily to their inhibitory (decoy) effect on pathogen binding to host cells in the small intestine [53]. Table 1 presents the name of the pathogens, which were found to be prevented by inhibitory/decoy activity of Sia components. Investigative reports on the effect of Sia specifically on porcine pathogens are very limited. Nevertheless, the microorganisms detailed in Table 1 or their counterparts are also found in pigs suggesting that Sia will resist pathogens and promote beneficial gut bacteria in pig.

### 4.3. Immune Function and Inflammation

Sia also plays an important role in immune function and its regulation. Inflammatory diseases can develop in early life and prevention or reduction of early inflammation may prevent the subsequent development of disease. Sia is present on the surface of monocytes and dendritic cells (DCs), as part of glycan structures and appears to be involved in the regulation of endocytosis and immune activation. During DCs maturation, the Sia composition on the cell surface changes resulting in an improved capacity for bacterial endocytosis and induction of a pro-inflammatory T-cell response [69,70]. Early T-cell activation has also been associated with increased Sia content on the T-cell surface [71]. This surface Sia is involved in the interaction of T cells with antigen-presenting cells (e.g., DCs, B cells), which express specific receptors for Sia, such as Sia-binding immunoglobulin-like lectins, known as Siglecs [72]. Many Siglecs also have inhibitory effects on immune cells and are therefore believed to be important for immune regulation [73]. Additionally, Sia has been shown to down regulate immune responses through its action on immunoglobulin G (IgG). Sialylated IgG binds to inhibit IgG receptors on DCs than activating them and this in turn down regulates the immune response [74].

Sialylation of compounds particularly sialylated oligosaccharides have been demonstrated to possess anti-inflammatory properties. For example, 3′-sialyllactose (3′-SL) has anti-inflammatory properties supported by the reduced levels of pro-inflammatory cytokines, such as IL-8 and TNF-alpha in CaCo-2 cells [75]. This effect was mediated via the enhanced expression of peptidoglycan recognition protein 3 (PGlyRP3), a pathogen recognition receptor shown previously to regulate inflammatory responses in vitro [75]. In a rat model of necrotizing enterocolitis (NEC), sialylated oligosaccharides supplementation prevented the occurrence as well as reduce the pathology of NEC and Sia was shown to be obligatory for this effect [76]. Collectively, these data suggest that Sia is an important building block for developing adequate immune and anti-inflammatory function, particularly in infancy.

### 4.4. Brain Development and Cognition

Neural cell membranes in mammals contain ~20-fold higher levels of Sia than other cell membranes, which indicates the key role of Sia-glycans in neural structure and function [77]. It has also been suggested that Sia is the actual receptor for neurotransmitters in the central nervous system [51]. Sia appears to be able to pass the blood-brain barrier [78,79]. Several animal studies (Table 2) suggest that oral Sia administration increases brain Sia content, as well as improving learning function.

## 5. Sialic Acid Concentration in Porcine Milk

Milk is a rich dietary source of Sia and the concentration varies based on species. Biologically for each animal species, mother’s milk is the best source of Sia for their offspring. For instance, human milk is a rich source of Sia for human infants, ranging from ~1.5 g/L in colostrum to 1 g/L in transition milk and 0.35 g/L in mature milk of full term mother at 3 months postpartum [86]. Preterm mother, however, contained 13–23% more Sia in the milk than that of full term mother during the first 3-months of lactation [86]. Sow’s milk is the major and only source of nutrients specific for newborn piglets. However, there is limited information on the concentrations of Sia in porcine milk over the course of lactation and if the mammary gland in different parity of sow produce different concentration of Sia. One of our study reported that the mean total concentration of Sia in the porcine milk is 1.24 g/L in colostrum, significantly reducing to 0.78 g/L in transition milk and further lowering to 0.35 g/L in mature milk at 20 day’s postpartum [87]. The dropping of total Sia concentration in milk with the progress of lactation may suggest a time-dependent reduction in the synthetic capacity of Sia and or an increased dilution of Sia with increased milk production as lactation progresses. Colostrum and mature milk from gilts (first parity) contains significantly lower concentrations of KDN and significantly higher concentrations of Neu5Ac than sow (multiple parity) milk (*p* < 0.05). [87]. The majority of Sia in porcine milk is conjugated to glycoprotein (41 to 46%), followed by oligosaccharides (31–42%) and gangliosides (12 to 28%), irrespective of the stage of lactation [87]. Neu5Ac is the major form of Sia (93–96%), followed by Neu5Gc (3–6%) and then KDN (1–2%) [87]. With the well-known fact that gilts are more likely to deliver low-birth weight piglets than sows [88], we hypothesize that dietary supplementation with Sia in gilts may improve the total milk Sia concentration and may possibly improve the condition of their litter. Further study is required to demonstrate our hypothesis. In addition, Mudd et al. [89] reported that the ratio between free and conjugated forms of Sia in porcine milk significantly altered over the course of lactation, with a decrease in the free Sia content and an increase in the conjugated form of Sia as the lactation progresses.

## 6. Sialic Acid in Different Organs of Pig

In addition to milk, animal derived food including egg, fish, red meat and dairy products are also a good source of Sia [90,91]. Recently we have demonstrated that Sia is detectable in various organs, including the brain, kidneys, heart, liver, spleen and lung in pig [14,91]. We also determined the developmental changes of free and conjugated Sia Neu5Ac, Neu5Gc and KDN in different organs of pig at 3, 38 and 180 days of age [90]. We demonstrated that tissue from 3 days-old piglets contained the highest level of total Sia (14.6 μmol/g protein) compared with other organs or age groups. During development, the level of total Sia showed a progressive developmentally-related decrease (44–79%) with age from postnatal age 3 to 180 days. Conjugated Neu5Ac was the major Sia in all tissues (61–84%) except spleen, where conjugated Neu5Gc was 2–4-fold higher than conjugated Neu5Ac at all three ages. Skeletal muscle contained the lowest concentration of Neu5Gc in fresh and cooked meat. KDN accounted for <5% of the total Sia in most organs, while free Kdn was the major Sia in skeletal muscle. Thus, Sia expression in pig is tissue and age specific. The high level of Neu5Gc in all organs compared to skeletal muscle is a potential risk factor suggesting that dietary consumption of organ meats should be discouraged in favour of muscle to protect against cancer, cardiovascular and other inflammatory diseases. In addition, of milk, Sia is present in the body fluids such as plasma, saliva, gastric juice, urine and tears, in the form of glycoproteins or as terminal sugars of oligosaccharide chains of mucins [79]. Different organs and body fluid express different concentration and form of Sia, which play important biological functions impact to pig health, disease and production.

## 7. Factors Affecting Sialic Acid Content of Milk

Although milk is one of the most abundant dietary sources of Sia, a systematic study to ascertain the factors that affect the concentration of Sia in milk is yet not available. However, many factors might influence Sia levels in milk. Inactivation or mutation of the sixty genes known to be involved in Sia biology [92] may result in abnormal biosynthesis and secretion of Sia from the mammary gland influencing the Sia content of milk. For instance, the missense mutation of GNE, the gene which encodes the enzyme responsible for synthesis of Sia, resulted in two genetic disorders of sialylation in humans [93,94,95,96].

Sia concentration in milk varies according to the species of the animal [86,87,97,98,99]. However, the different level of milk [99] Sia in different species implies that the capacity of Sia synthesis by animals during lactation, as well as, physiological requirement by their offspring are species specific. In addition, nutrient intake and health status of lactating animals might influence Sia concentration in their milk during lactation. Additionally, milk Sia concentration vary over the course of lactation, e.g., the highest concentration of Sia in colostrum is irrespective of the species studied [86,87,97,98,99]. Asakuma et al. [100] demonstrated that total Sia concentration in milk was significantly higher in pasture grazed cows compared to grass silage fed cows, suggesting the influence of feed on Sia levels in milk. Although there is no significant difference in the total Sia content of milk collected in different seasons in dairy cow, the concentrations of gangliosides, which are Sia-containing glycolipids, are higher in bovine, caprine and ovine milks in the fall compared to other seasons of the year [101,102]. These results imply that Sia concentration in animal milk might be influenced by seasons. Sia levels in milk may also be influenced by breed and age of animal, but further studies are warranted to confirm.

## 8. Sialylated Milk Oligosaccharides in Porcine Milk

To date, there are limited reports that have characterised the Sia-MOS’s in porcine milk. About 119 porcine milk oligosaccharides (PMOs) have been characterized [103], of which 25 are very new for the PMO literature [103]. Total concentration of PMOs is 11.85–12.19 g/L and 6.82–6.98 g/L in colostrum and mature milk, respectively [104]. Porcine milk contains both acidic (Sia-MOS’s) and neutral oligosaccharides. Sia-MOS’s is the most predominant type of oligosaccharide in sows (56.2–78.17%) and gilts (61.12–76.90%) [103]. So far 43 Sia-MOS’s structure have been identified which accounts for 16–80% of the oligosaccharides structures in pig milk [103]. 3′-SL is the major PMOs with concentration of 5.03–20.98 g/L in colostrum representing 68–71% of total PMOs [103]. In porcine milk, the total proportion of Sia-MOS’s declines from 77–80% at farrowing to 60% in early lactation (days 4–7) to approximately 40–49% in late lactation (day 24) [105]. The proportion of Neu5Ac and Neu5Gc in PMOs is 99.9/0.1 [105]. The total number of PMOs in gilt and sow almost similar in colostrum (53 vs. 54), but is different in transitional (48 vs. 53) and mature milk (41 vs. 47) [103].

## 9. Sialylated Glycoprotein Lactoferrin

Lactoferrin (LF) was an 80 kD iron binding glycoprotein containing Sia residues attached to the *N*-linked glycan chains [106]. In human milk, this glycoprotein can harbor 1-4 Sia residues [107]. Glycosylation of LF has been extensively studied in humans but not in animals. The *N*-glycans of human LF (hLF) are highly branched and sialylated and contain complex fucosylated complex structures [108]. In adult mammalian species, LF is produced by mucosal epithelial cells and is found in various mucosal secretions, including tears, saliva, vaginal fluids, semen [109], nasal and bronchial secretions, bile, gastrointestinal fluids and urine [110] and milk. However, the highest levels of LF are detected in colostrum and milk of the human, making it the second most abundant protein in milk [111], after caseins.

LF has been found in the milk of numerous mammals, with the amino acid sequence known for human, pig, horse, buffalo, cow, goat, camel, sheep and mouse LF [112]. The concentration of LF in milk and colostrum varies amongst species. Yang, Zhu, Liu, Chen, Gan, Troy II and Wang [111] concluded that the LF concentration in humans is ~9.7 mg/mL in colostrum and 2–3 mg/mL in mature milk. Bovine mature milk contains approximately one tenth the amount of LF as human milk, ranging from 0.03–0.1 mg/mL [111]. Literature on LF levels in the milk of pigs is very limited. Elliot et al. [113] reported that LF levels averaged 1.1–1.3 mg/mL at the time of farrowing, with concentrations sharply declining during the first week of lactation in Yorkshire sows.

Recently, we compared milk lactoferrin concentration of primiparous and multiparous sows during lactation using uHPLC method. We found porcine milk contained significant levels of LF with the highest concentration in colostrum (9.68 mg/mL), which decreased by ~62% and ~67% in transitional (3.69 mg/mL) and mature milk (3.22 mg/mL), respectively. The mature gilt milk contained a 22% higher concentration of LF (3.67 mg/mL) compared with sow milk (<0.05). A possible explanation for this finding can be attributed to a compensatory response by gilts, wherein they produce higher levels of LF in mature milk to help compensate for milk volume deficiencies, thereby supporting the proper growth and development of their piglets. Interestingly, the breed line had an overall significant effect on the LF content of porcine milk, but not the litter size [114]. We concluded that LF is an important constituent of pig milk that might contribute to the optimum growth and development of piglets.

LF has several physiological functions, including antianemic, anti-inflammatory, antimicrobial, immunomodulatory, antioxidant and anticancer activities. LF modulates immune function and its involvement as a first line of host’s defense response against bacteria, viruses and fungi [115,116] and parasites [117] and it is considered to be a mediator linking innate and adaptive immune responses. A meta-analysis of data from twelve randomized clinical trials demonstrated that enteral LF supplementation significantly prevented sepsis and necrotising enterocolitis in preterm infants [118]. Our research team has completed a series of studies to demonstrate the role of LF on neurodevelopment, neuroprotection and gut development in piglet. A summary of key studies on the effects of oral supplementation of LF in pigs shown in Table 3.

## 10. Sialylated Glycolipid Ganglioside

Gangliosides (GAs) belong to the large family of complex lipids (sphingolipid) and usually contain a ceramide (made up of sphingosine and fatty acids) moiety attached to Sia containing complex oligosaccharides. Neu5Ac is the most common form of Sia found in the hydrophilic oligosaccharide head group. Less commonly Neu5Gc may be found [129]. Gangliosides were first isolated from ganglion cell of neural tissues [130]. Since then, GAs have been found in almost all tissues and are present in considerably higher concentrations in neural tissues and extra-neural organs, such as the lung, spleen and gut. They are also found in biological fluids such as blood and milk localized in the fat globule membrane in milk [131,132]. The variety of GAs depends mainly on the extent of glycosylation, more specifically of sialylation and the diversity of glycosidic linkages which give rise to different classes of GAs.

Gangliosides GD3 and GM3 are the major components of GAs fraction in human milk at the concentration ranging from as low as 1.7 mg/L [97] to as high as 23.8 mg/L [133]. Human colostrum, bovine milk and infant formulas contain GD3 as the main GAs representing 50%, 60% and 80% of the total amount of GAs, respectively. The GAs concentration in milk is influenced by the stage of lactation. From transitional to mature milk, the GM3 concentration rises up to 50% in human milk, while the GD3 concentration declines [134]. Different extraction and analytical methodologies for GA analysis, population demographics, diet, time of sampling and study protocol influence the variation in milk GA concentration [135]. The concentration of GAs in the colostrum and mature milk of cow is two times lower than that of human milk [134].

To date, the concentrations of total gangliosides in porcine milk has not been researched upon. However, we demonstrated that the total Sia conjugated to gangliosides in the colostrum, transition milk and mature milk were 148.98 ± 23.9, 214.24 ± 15.2 and 70.04 ± 9.5 mg/mL respectively [92]. Our results imply developmental changes in the sialylation of gangliosides in the porcine milk over the course of lactation. In addition, the total amounts of Sia conjugated to gangliosides did not significantly differ between gilts and sows [87]. Given that gangliosides, a Sia-binding glycolipid, play critical roles in the normal physiology and functioning of the body, it is imperative that these sialylated glycolipids are present in adequate amounts in pig’s milk to serve as one of the major source of Sia to the offspring

The biological functions of GAs (both endogenously synthesised and of dietary origin) have been extensively discussed in the literature [136,137,138]. GAs play vital roles in brain development, establishment of neuronal network, memory formation and synaptic and transmembrane signal transduction. In addition, GAs are important for proper cell adhesion, proliferation and differentiation. Gurnida et al. [139] reported that nutritional supplementation with a ganglioside-rich milk lipid preparation seems to have beneficial effects on cognitive development in healthy infants at 0–6 months of age. They have also been associated with modification of intestinal microflora (protection against enteric pathogen and proliferation of *Bifidobacterium* sp.) and regulation of the immune system and gut maturation of infants.

## 11. Conclusions

Porcine milk is a rich source of Sia and consequently porcine offspring require a large amount of Sia to support their growth and development. Sia plays an important role in ameliorating the pathogenicity of bacteria and other infectious microorganisms. However, the compositional differences in Sia moieties between sow and gilt milk may influence growth, development and health of their progeny, but clearly this is an area that requires further intensive investigation. On the other hand, supplementation of gilts with sialylated glycoprotein, LF during gestation and lactation markedly improved pig production by increasing production of milk, thereby enhancing growth of piglets to weaning and by improving the immune status of gilts and their offspring. Thus, after a large-scale confirmatory study, inclusion of Sia or Sia containing oligosaccharide, LF in gilt and sow feed is recommended to increase the efficiency and therefore profitability of pig production in commercial farming systems. In addition, a sow milk replacer can be developed based on the Sia concentration in porcine milk for IUGR piglets, or for the piglets with dams suffering from agalactia, a common problem in pig production systems worldwide.

## Figures and Tables

**Figure 1 animals-11-02318-f001:**
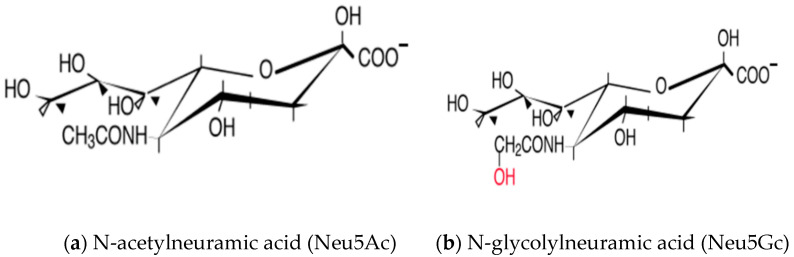
The structures of three most common Sialic Acids. (**a**) *N*-acetylneuramic acid (Neu5Ac), (**b**) *N*-acetylneuramic acid (Neu5Ac) and (**c**) Ketodeoxynonulosonic acid (KDN).

**Figure 2 animals-11-02318-f002:**
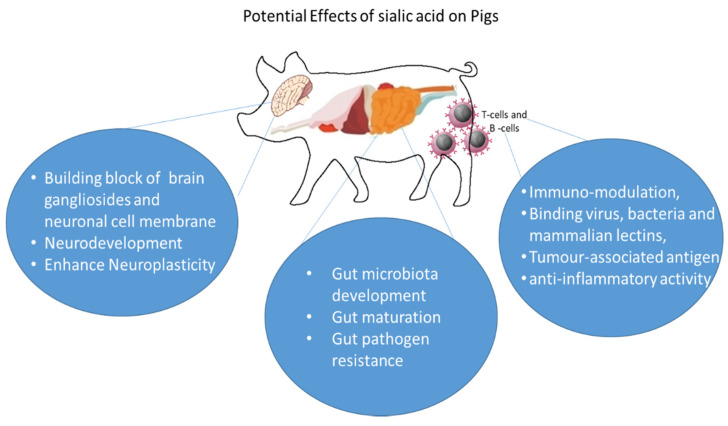
Potential effects of sialic acid on Pigs.

**Table 1 animals-11-02318-t001:** Microorganisms against which sialic acid exhibits inhibitory/decoy activity.

Study Model	Pathogen	Reference
In vitro	*Group B Streptococcus*	[54]
In vitro	*Salmonella*	[55,56]
	*Various E. coli* strains (S-fimbriated strains in particular)	[55,56,57,58]
	*Vibrio cholera*	[55]
	*Helicobacter pylori*	[55,59,60,61]
	*Campylobacter* *Jejuni*	[55]
	Rotavirus	[62,63]
Mice	*H. pylori*	[64]
	Rotavirus	[53,63,65]
	cholera toxin (specific for SL)	[65]
Rhesus monkey	*H. pylori*	[66]
Human infant	Rotavirus	[67]
Pig	Rotavirus	[68]

**Table 2 animals-11-02318-t002:** A summary of studies exploring the in vivo effect of oral Sia supplementation in brain development and cognition in rodents and piglets.

Animal	Dose of Sia	Duration of Treatment	Effect	References
Piglet: 3 days old, male (average weight 1.5–2.4 kg)	Sia @40, 85, 180 or 240 mg/Kg body wt/day	35 days	Increased learning and memory function when exposed to eight arm radial maze learning tests, increased brain sialylated glycoprotein and also increased levels of mRNA expression of uridine-diphospho-*N*-acetlylglucosamine-2-epimerase, a key enzyme in the biosynthetic pathway of Sia, in the brain and liver.	[80]
Rat: 17 days old	Sia @20, 40 or 60 and 240 mg/Kg body wt/day	16 days	Increased cortical ganglioside Sia content.	[81]
Rat: 2 years old	Sia @ 0.8 g/100 g diet	3 weeks	Sia feeding in aged rats normalized brain gangliosides Sia levels to the levels measured in young rats.	[82]
Rat: 9 weeks old (average weight 300 g)	Galactosylated Sia or SL @ 1% of diet	2 weeks	Improved learning ability in a swimming learning test, which was associated with increased Sia and ganglioside content of the brain.	[83]
Rat: 12 weeks old pregnant rat (fed with n-3 fatty acid-deficient diet for 3 weeks before mating)	Sia @ 1% of water ad libitum	Throughout pregnancy and lactation	Improved the recognition index of novel object recognition test in rat pups after weaning. Increased the level of dimethylacetal (from plasmalogen) in the cerebral cortex of weaned pups. There was no effect in the total Sia content in the brain cortex or hippocampus of rat pups.	[84]
Rat: 14 days	Sia @ 20 mg/Kg body wt/day	8 days	Intraperitoneally or orally administered Sia in rat pups resulted in increased cerebral and cerebellar ganglioside concentration.	[85]

**Table 3 animals-11-02318-t003:** A summary of key studies on the effects of oral supplementation of LF in pigs.

Age of Piglet	Type and Dose of LF	Duration of Treatment	Effect	References
Postnatal piglet (3 days)	bLF @ 155 mg/Kg body wt/day and 385 mg/Kg body wt/day	35 days	(1) Enhanced learning and long-term memory are evident in low dose compared to the high dose. (2) Lower dose upregulates genes and functions correlated with neurodevelopment and cognition, while the higher dose regulates cellular processes for neuroprotection. (3) Expression of BDNF genes and proteins is higher with both concentrations, while genes regulating BDNF signalling responds more to the lower dose. (4) Lower dose modulates genes in the five highest networks associated with cellularity and neuro-cognition, while the prevention of neurodevelopmental and neurological pathologies is associated with the higher dose.	Chen et al. [119]
Primiparous sows (gilts) (average weight 151.8 kg ± 15.1 kg)	LF top dressed @1 g/day	Day 1 post mating until 21 days post lactation period (total ~135 days)	(1) Increased milk production over the course of lactation. (2) Increased body weight gain of their piglets during the first 19 days of life. (3) Increased the concentration of serum IgA in gilt and serum sIgA in piglets. (4) Tended to increase pregnancy rate, litter size and birth weight, number of piglets born alive and decrease the number of dead and intrauterine growth restriction (IUGR) piglets.	Jahan et al. [120]
Postnatal piglet (3 days)	bLF @ 155 mg/Kg body wt/day and 385 mg/Kg body wt/day	35 days	(1) Upregulates mRNA level of gene encoding brain derived neurotropic factor (BDNF) which has, neurotrophic and neuroprotective role in the development and regeneration of CNS and ubiquitin thio-carboxy terminal hydrolase 1 (UCHL1), an enzyme in neural cell. (2) Increases intestinal alkaline phosphatase (IAP) activity, an enzyme which regulates gut h omeostasis. (3) Increases the length of jejunal crypts. (4) Decreases colon microbiota of *E. coli*. (5) Reduces incidence of early weaning diarrhoea.	Yang et al. [111]
Postnatal piglet (3 days)	bLF @ 155 mg/Kg body wt/day	35 days	(1) Upregulates BDNF signalling pathway and the expression of polysialic acid (a marker of neuroplasticity) associated with neurodevelopment and cognition. (2) Enhances the cognitive function and learning ability of piglets.	Chen et al. [121]
Newborn piglet from day 1 of age	bLF @ 155 mg/Kg body wt/day and 385 mg/Kg body wt/day	14 days	(1) Increases intestinal cell proliferation along with enhanced crypt depth, crypt area and villus length.	Reznikov et al. [122]
Weaned piglet (21 days)	Recombinant bovine lactoferrampin-lactoferricin (LFA-LFC-Lactoferrin derivative) @ 100 mg/Kg, 3 times/day	21 days	Increases the intestinal microbiota by increasing the *Lactobacillus* and *Bifidobacterium* in the chime of GIT.	Tang et al. [123]
Weaned piglet (21 days)	Lactoferrin derivative LFA-LFC @ 100 mg/Kg, 3 times/day	21 days	(1) Enhances growth performance. (2) Promotes recovery from diarrhoea. (3) Improves antioxidant status. (4) Increases serum IgA, IgG and IgM level. (5) Decreases intestinal microbial content of *E. coli* and increases that of *Lactobacillus* and *Bifidobacterium*. (6) Promotes the development of villus- crypt architecture.	Tang et al. [124]
Weaned female piglet (28 days)	bLF @ 1 g//Kg body wt/day	30 days	(1) Induces peripheral and spleen lymphocyte proliferation. (2) Increases serum IgA, IgG and IgM, complement 4 (C_4_) and IL-2 level. (3) Increases serum iron level.	Shan et al. [125]
Female piglet (35 days)	bLF @ 1250 or 2500 mg/Kg body wt/day	30 days	(1) Increases growth performance. (2) Enhances antioxidant enzyme activity and mRNA level. (3) Decreases lipid peroxidation.	Wang et al. [126]
Weaned piglet	1 g/kg body wt/day	15 days	(1) Improves growth performance. (2) Increases intestinal villus height and crypt depth. (3) Upregulates the mRNA expression of 39 residue proline-arginine-rich peptide (PR-39) and protegrine 1 in bone which are indicative of improved immune function.	Wang et al. [127]
Weaned piglet	1 g/kg body wt/day	30 days	(1) Improves growth performance. (2) Decreases the incidence of diarrhoea. (3) Decreases the total viable count of *E. coli* and enriched *Lactobacillus* and *Bifidobacterium* in the intestine. (4) Increases villus height and decreases crypt depth.	Wang et al. [128]

## Data Availability

Not applicable.

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
