# Peer review of "The Potential for Sialic Acid and Sialylated Glycoconjugates as Feed Additives to Enhance Pig Health and Production"

_animals, 2021, doi:10.3390/ani11082318_

Round 1
Reviewer 1 Report
Dear authors,
Congratulations for this very interesting review.
Author Response
Many thanks for your inspirational comments about our manuscript. We really appreciate that.
Reviewer 2 Report
In pig farming new feed additions are sought that can enhance animal performance by improving growth and immune status. Therefore, the manuscript presented for review is a valuable source of information on the subject the potential nutritional significance of sialic acid (Sia) and glycoconjugates (Sia-GC’s) for pig health and nutrition. Information presented in the article will assist in optimizing the composition of sow/gilt milk replacers designed to increases the survival of IUGR piglets or piglets with dams suffering from agalactia. I believe that the article should be published in the journal Animals because it is a valuable, reliable source of information on the topic contained in the title.
Author Response
Au: Thank you very much for your positive comments about our review. We also believe that this review is a valuable and reliable source of information on the topic. Thanks again!
Reviewer 3 Report
In general there is a lot of good information in the review (make sure it is put in as a review, it shows up on my end as an original research article), but there is maybe too broad of information. By this I mean that there is a lot information in the body of text that could be tied to pig health but the link is not provided by the authors. One other general preference of mine is fewer abbreviations make articles easier to read.
I do not see any examples of sialylated gycoconjugates as feed additives. I see data about what is happening in nature but would be interested to a feed additive source. With this I think either the title need modification or that data needs to be added. At this juncture the title seems to be a hypothesis not a statement. A title that is more fitting would be "The potential impact of the heterogeneity of sialic acids in nature and variation during infant development on pig health and production."
Please fix all of the places with multiple spaces and random letters, I pointed some out below but not all of them.
Line 16: Change to sows
Line 25: challenges would be more appropriate than limitations
Line 54: add; is one of the most
Line 66-69: Re-word and break into multiple sentences
Line 70: Is cause the correct word? I would think that immunodeficiency would be associated with IUGR piglets, not caused by it.
Line 76 and 77: why give these 3 examples and not talk about the us or china or brazil?
Line 102: spelling; Saliva
Line 106: remove the extra space
Tie sections 3 and 4 together, as a reader it is not clear what impact the variation has on the biological functions.
Line 157; extra space
Line 228: extra space
Line 326: random s, no period
Line 357: That citation needs consolidated
Line 366-375: repeated from above
Table 4: Make citations consistent
Line 413: citations should be in the same set of brackets
Line 455-458: This seems to run on and I think it needs to be referenced
Author Response
In general there is a lot of good information in the review (make sure it is put in as a review, it shows up on my end as an original research article), but there is maybe too broad of information. By this I mean that there is a lot information in the body of text that could be tied to pig health but the link is not provided by the authors. One other general preference of mine is fewer abbreviations make articles easier to read.
Au: Thank you for your nice comments on our manuscript. All the abbreviations used in this manuscript has been splined out in its first use to make it easy to read.
I do not see any examples of sialylated gycoconjugates as feed additives. I see data about what is happening in nature but would be interested to a feed additive source. With this I think either the title need modification or that data needs to be added. At this juncture the title seems to be a hypothesis not a statement. A title that is more fitting would be "The potential impact of the heterogeneity of sialic acids in nature and variation during infant development on pig health and production."
Au: Although the publications of sialylated glycoconjugates used as feed additive are limited, our previous studies have shown using sialylated milk oligosaccharides as feeds additive for brain and gut development of piglet in postnatal life (PPMID: 31746283, PMID: 31161424.). In addition, two research projects from our group reported that two glycoprotein conjugated sialic acids: lactoferrin and casein-glycomacropeptide as feed additive have beneficial role on pig production for pregnant sow and brain and gut development for newborn piglet. All these studies have been citated in the review. Therefore, this review focuses on current knowledge of the importance of different source of sialic acid as feed additive for pig health.
We really thank and respect your suggestion to change the title, however, we believe that our current title ‘the potential for sialic acid and sialylated glycoconjugates as feed additives to enhance pig health and production” is more suitable to the point of the script.
Please fix all of the places with multiple spaces and random letters, I pointed some out below but not all of them.
Au: We really appreciate your careful review of our manuscript. All such errors regarding the space and random letters have been corrected throughout manuscript.
Line 16: Change to sows
Au: Thanks, corrected.
Line 25: challenges would be more appropriate than limitations
Au: The word ‘limitations’ has been replaced by ‘challenges.’
Line 54: add; is one of the most
Au: Thanks, done.
Line 66-69: Re-word and break into multiple sentences
Au: Thanks for your suggestion. The sentence has been reworded and has been broken down into sentences.
Line 70: Is cause the correct word? I would think that immunodeficiency would be associated with IUGR piglets, not caused by it.
Au: The word “cause’ has been replaced with ‘be associated with’.
Line 76 and 77: why give these 3 examples and not talk about the us or china or brazil?
Au: Thanks for your query. It has been mentioned in the earlier sentence that ‘the mean piglet Pre-weaning mortality (PWM) rate in commercial pig herds ranges between 10% and 20% in major pig-producing countries”. However, the more specific data is available about European Union (EU), the Philippines, and Thailand only, therefore these countries has been included.
Line 102: spelling; Saliva
Au: The spelling has been corrected.
Line 106: remove the extra space
Au: Corrected
Tie sections 3 and 4 together, as a reader it is not clear what impact the variation has on the biological functions.
Au: Thanks for your suggestion. Different source of sialic acid in feeds might have different biological function, therefore it will be better to summarize different sources of sialic acid and biological function separately, which will be easier for the readers to follow.
Line 157; extra space
Au: Corrected.
Line 228: extra space
Au: Corrected.
Line 326: random s, no period
Au: Corrected.
Line 357: That citation needs consolidated
Au: The citation has been corrected.
Line 366-375: repeated from above
Au: Sorry for the mistake. The repeated section has been deleted. Thanks!
Table 4: Make citations consistent
Au: The citations in table 4 is consistent now.
Line 413: citations should be in the same set of brackets
Au: Corrected
Line 455-458: This seems to run on and I think it needs to be referenced
Au: Thanks for your suggestion. But this is our own conclusion to this review and thus cannot be referenced.
This manuscript is a resubmission of an earlier submission. The following is a list of the peer review reports and author responses from that submission.
Round 1
Reviewer 1 Report
The review is focused on pig production, but information on pigs is scarce (18 references out of 142). Most part of the article does not focus on pigs and could be applied to any other animal especie.
I understand that there is not much scientific information about pigs, but I would have expected a more deep discussion on the effect of Sialic acid compounds on mortality, growth, performance, and so on. Biochemical and physiological approach (based mostly on information from human orientated studies) lack novelty.
Table 4. PLease, provide reference number for Jahan publication
Ln 346-359. I believe there is a duplicity in this paragraph and that on line 376-382
Reviewer 2 Report
Manuscript ID: Animals-1236633
“Nutritional significance of sialic acid and sialylated glycoconjugates in pig health and production”
This review set out to examine the nutritional role of sialic acid and sialylated glycoconjugates in swine.
GENERAL COMMENTS:
The manuscript suffers from important limitations. Authors specify (see title) that the review is focused on swine. However, most of the references are not related to swine; in addition they need to be updated. The paper could be more efficiently written.
SPECIFIC COMMENTS:
Simply Summary: it is omitted. Please, remove the instructions to authors and add a simply summary.
L.56: please update the reference 1: Anderson and Parker 1976.
L.145: please insert the number (5) and the title of the section. “5.1 Development of gut microbiota” is a subsection.
Subsection 5.1, 5.2, 5.3 and 5.4: Please, check and change references: Almost all of the references are specific to humans and rats.
Section 9: Free and conjugated sialic acids Neu5Ac, Neu5Gc and KDN in different organs of pig
L.328-331: please change the reference [80]: it is not specific to swine.
Section 10: Sialylated glycoprotein Lactoferrin
L347-369: please check the sentences (some sentences are repeated) and the format of references.
Table 2: please remove the table. Data are almost all referred to the rats.
Reviewer 3 Report
Dear authors,
Many thanks for this review which I found very interesting, as a reader first. I believe that your work is meritorious of being published, but it can be improved.
After reading the review I found some weak points which should be recovered before publication.
One is that the review is unbalanced across the different topics you dealt with. In fact, you report different aspects related to farming system, then to health problems in pigs, then to a very extensive biochemistry description of Sias and then again some comparative aspects. On the whole, I really feel that the distribution of topics within the review is somewhat unbalanced and not enough focused on what is the real meaning you aimed to convey through this review. Some citations appear to be more related to strictly biological aspects of Sias inti a specialistic approach, and as a general speech, others do not relate strictly to pork productions. Moreoever, I feel that the real role for healthy pig raising (beyond production) is failing to point to the core in your description, as I expected from reading the title of the review.
Minor concern:
The simple summary is totally missing and this may help to focus more on the core of the review.
Many thanks! Please, balance and focus the review.